# Characteristic of Concurrent Uterine Lipoleiomyoma and Hemangioma by Algorithm of Candidate Biomarkers for Uterine Mesenchymal Tumor

**DOI:** 10.3390/diagnostics12102468

**Published:** 2022-10-12

**Authors:** Shoko Nishikawa, Takuma Hayashi, Yasuaki Amano, Nobuo Yaegashi, Kaoru Abiko, Ikuo Konishi

**Affiliations:** 1Department of Obstetrics and Gynecology, National Hospital Organization Kyoto Medical Center, Kyoto 606-8501, Japan; 2Section of Cancer Medicine, National Hospital Organization Kyoto Medical Center, Kyoto 612-8555, Japan; 3Seeds Development and Research Infrastructure Division, Japan Agency for Medical Research and Development (AMED), Tokyo 100-0004, Japan; 4Department of Obstetrics and Gynecology, Tohoku University School of Medicine, Miyagi 980-8575, Japan; 5Department of Obstetrics and Gynecology, Kyoto University School of Medicine, Kyoto 606-8501, Japan

**Keywords:** lipoleiomyoma, hemangioma, bleeding, uterine leiomyoma, uterine leiomyosarcoma

## Abstract

(1) Background/Aim: In clinical practice, uterine lipoleiomyomas are variants of uterine leiomyomas that are often found incidentally and do not require surgical treatment unless the patient is symptomatic. Therefore, these should be clinically differentiated from lesions that need surgical treatment. Conversely, hemangiomas, or blood vessel benign tumors, rarely develop in the uterus; however, many clinical complications such as abdominal pain and excessive vaginal bleeding result from a uterine hemangioma. Hemangiomas can occur at any age and primarily affect pregnant women. (2) Materials and Methods: The oncological properties of uterine lipoleiomyoma and hemangioma in adults were investigated using molecular pathological examination on tissue excised from patients with a uterine tumor. (3) Results: Through molecular pathological studies, which included potential biomarkers for uterine mesenchymal tumors, a differential diagnosis was established for a case of mesenchymal tumor. Herein, we report a 54-year-old non-pregnant woman who presented with vaginal bleeding and underwent hysterectomy after detection of a 140 × 100 mm intramural mass diagnosed as a concurrent uterine hemangioma and lipoleiomyoma after molecular histopathologic examinations. (4) Conclusion: As far as we know, our case is the first patient of concurrent uterine hemangioma and lipoleiomyoma. Hence, the possibility of several types of mesenchymal tumors must be considered in the differential diagnosis of patients with abnormal vaginal bleeding. As such, molecular pathological examination and close monitoring of the MRI results should be conducted by medical staff while considering the patient’s desire for pregnancy, including surgical treatment options for uterine hemangioma.

## 1. Introduction

Among all gynecologic tumors, uterine leiomyoma is the most common benign gynecological tumor. The majority of uterine leiomyomas are found in women in their 40s and 50s [1]. By age, the incidence of uterine leiomyoma is thought to occur in 20–30% of women aged 30 years and over and in 70% of women aged 40 years and over [2,3]. Women of sexual maturity are the most affected population with clinically relevant uterine leiomyomas. Moreover, the incidence of uterine leiomyoma is significantly lower in women under the age of 18 years [2]. Additionally, uterine leiomyoma observed before menopause has been shown to shrink after menopause [3]. Furthermore, a rapid growth of uterine leiomyomas has been demonstrated in pregnant women and women taking oral contraceptives. Based on these clinical findings, a uterine leiomyoma is thought to be female hormone-dependent [2,3].

In many cases of uterine leiomyomas, spindle-shaped smooth muscle cells proliferate in mutually orthogonal fascicles or whorls. Uterine leiomyoma cell nuclei are uniform and elongated, with round to blunt cell ends. In uterine smooth muscle tumor cell size, there is almost no difference, or the difference is extremely mild. Uterine leiomyomas have minimal or no increase in chromatin and have very small or non-existent nucleoli. Therefore, in uterine leiomyoma, nuclear atypia is absent. Uterine leiomyomas with mixed adipocytes are classified as lipoleiomyoma. Uterine leiomyoma with a conspicuous fatty component is referred to as a lipoleiomyoma [3].

Hemangioma, on the contrary, refers to a benign tumor formed by the dilation or proliferation of blood vessels. Medically, hemangiomas depict vascular endothelial cell proliferation, while vascular malformations are not characterized by proliferation. Hemangiomas and vascular malformations on the body surface and soft tissue are often commonly referred to as “hemangioma”. However, hemangioma and vascular malformation are different diseases according to the International Society for the Study of Vascular Anomalies (ISSVA) classification, which is internationally standardized and advocated by the International Society for the Treatment of Hemangiomas and Vascular Malformations [4]. Hemangiomas include infantile hemangiomas and tufted hemangiomas. Vascular malformations include port-wine nevus (simple hemangiomas), cavernous hemangiomas, and arteriovenous malformations. Uterine hemangiomas are rare benign vascular tumors that can cause abnormal vaginal bleeding in different age groups. Patients with these tumors present with menorrhagia or pregnancy-related complications. Magnetic resonance imaging (MRI) is usually utilized for the differential diagnosis between hemangiomas and other diseases; however, only a few studies elucidate its detection ability. Compared with other imaging tests, MRI has a high-detection capability for uterine hemangiomas, and comprehensive reports on hemangioma reveal the detection rate by MRI. In general, MRI imaging is better at detecting hemangiomas than ultrasound or computed tomography (CT) imaging [5,6]. In particular, the detection rate of hemangiomas on T2-weighted imaging is 93–100% [6,7]. Incidental findings of hemangiomas using MRI are often encountered in daily clinical practice, and establishing a correct diagnosis is paramount.

A 53-year-old non-pregnant woman with significant vaginal bleeding presented to our medical facility. An outpatient MRI imaging study revealed a mass measuring 140 × 100 mm, suggestive of uterine lipoleiomyoma. Surgical treatment was performed, and the tissue was sent for pathological examination, which revealed a concurrent uterine lipoleiomyoma and hemangioma. We report a case of concurrent uterine lipoleiomyoma and hemangioma. To our knowledge, this is the first case of concurrent uterine hemangioma and intramural lipoleiomyoma. In the differential diagnosis of patients with abnormal vaginal bleeding, it is necessary to consider the possibility of such uterine hemangiomas.

## 2. Materials and Methods

### 2.1. Immuno-Histochemistry (IHC)

Our clinical research group carried out IHC staining for caveolin 1, cyclin B, cyclin E1, large multi-functional peptidase 2/β1i (LMP2/β1i), Ki-67/MIB1, and cluster of differentiation 31 (CD31) using serial human uterine mesenchymal tumor sections obtained from patients with uterine mesenchymal tumors (see clinical situation history in Appendix A). Our clinical research group purchased a monoclonal antibody against cyclin E1 (CCNE1/2460) from Abcam (Cambridge Biomedical Campus, Cambridge, UK) and a monoclonal antibody for Ki-67 (clone MIB-1) from Dako Denmark A/S (DK-2600, Glostrup, Denmark). Moreover, our clinical research group purchased monoclonal antibodies against CD31 (GTX130274) from GeneTex, Inc. (Irvine, CA, USA) and purchased the appropriate monoclonal antibodies for caveolin 1, cyclin B1, and LMP2/β1i from Santa Cruz Biotechnology Inc. (Santa Cruz, CA, USA). We carried out IHC experiment using the avidin–biotin complex method as described previously.

We incubated the tissue sections with a biotinylated secondary antibody (Dako, DK-2600, Glostrup, Denmark) conjugated with the streptavidin complex (Dako). The completed reactions were developed using 3,39-diaminobenzidine, and our clinical research group counterstained the slides with hematoxylin. We used normal myometrial tissues in the specimens as positive controls. The negative controls consisted of tissue sections incubated with normal rabbit immunoglobulin (IgG) instead of the primary antibody. Shinshu University (approval no. M192) and the National Hospital Organization Kyoto Medical Center approved the experiments according to internal guidelines (approval no. KMC R02-0702). The expression of cyclin E and Ki-67/MIB was indicated by brown 3,3′-diaminobenzidine (DAB) tetrahydrochloride staining. Our clinical research group used normal rabbit or mouse antiserum as a negative control for the primary antibody. The entire brown DAB tetrahydrochloride-stained tissue was scanned using a BZ-X800 digital microscope (Keyence, Osaka, Japan). The expression of cyclin E and Ki-67 was indicated by brown dots.

Our clinical research group carried out IHC staining for CD31 on sections from the excised tissue. Briefly, we incubated the tumor tissue sections with the appropriate primary antibodies at 4 °C overnight. We used a mouse monoclonal antibody to CD31 (1:200) as the primary antibody. In addition, we purchased a monoclonal antibody for CD31 (clone GTX130274) from GeneTex, Inc. (Irvine, CA, USA). Following incubation with an Alexa Fluor^®^ 488-conjugated anti-mouse IgG (1:200; Invitrogen, Waltham, MA, USA) as second antibody, the sections were washed, cover-slipped with mounting medium and 40,6-diamidino-2-phenylindole (Vectashield; Vector Laboratories, Burlingame, CA, USA), and visualized using confocal microscopy (Leica TCS SP8, Wetzlar, Germany). The clinical research experiments with human tissues were conducted at the National Hospital Organization Kyoto Medical Center in accordance with the institutional guidelines (approval no. NHO H31-02).

The photographs of uterine leiomyoma and uterine leiomyosarcoma tumor tissues subjected to immune-histochemical staining shown in the Results section were taken of a uterine tumor excised from a patient suspected of developing uterine leiomyosarcoma by our medical team. In many patients, uterine leiomyosarcoma co-occurs with uterine leiomyoma. Therefore, uterine leiomyoma and uterine leiomyosarcoma tumor tissues were surgically excised from the same patient.

### 2.2. Ethical Approval and Consent to Participate

The Central Ethics Review Board of the National Hospital Organization Headquarters in Japan (Meguro, Tokyo, Japan) and Shinshu University School of Medicine (Matsumoto, Nagano, Japan) reviewed and approved this study. Ethical approval was obtained on 17 August 2019 with the code NHO H31-02. The authors attended educational lectures on medical ethics in 2020 and 2021, which were supervised by the Japanese government. The completion numbers for the authors are AP0000151756, AP0000151757, AP0000151769, and AP000351128. As this research was considered to be clinical research, consent to participate was required. After briefing regarding the clinical study and approval of the research contents, the participants signed an informed consent form.

Details of the materials and methods are indicated in the Appendix A.

## 3. Results

Our Case

A 53-year-old non-pregnant woman was referred to our medical facility from a nearby general medical institution for further examination due to a pelvic mass found on transabdominal ultrasonography. Transvaginal ultrasonography revealed a well-circumscribed, bright, solid mass measuring 100 × 140 mm in diameter in the corpus of the uterus. Internal blood flow on Doppler and absence of posterior echo attenuation were depicted by the mass. Pelvic MRI T1-weighted fat suppression images showed signal suppression in the same region. In the gadolinium contrast images, no contrast enhancement effect was observed. A contrast-enhanced MRI T2 image showed that the tumor wall was isointense with the myometrium, and a beak sign in the tumor capsule and flow void findings, initially thought to be uterine artery branches, were observed (Appendix A). Thus, the mass in the uterine corpus was determined to be a large, fat-containing tumor derived from the uterus. On blood tests, tumor markers, and pathological examination by endometrial aspiration biopsy and pathological diagnosis by cervical cytology were negative. These findings suggest that the mass may have been a lipoleiomyoma originating from the uterine corpus. In clinical studies to date, the characteristic imaging findings suggest the possibility of lipoleiomyoma pre-operatively. However, in clinical practice, it is necessary to differentiate and exclude other malignant tumors such as well-differentiated liposarcoma and malignant transformation of ovarian mature cystic teratoma.

Surgical pathological findings using the excised tissue revealed multiple uterine leiomyoma-like nodules in the uterine corpus tissue. The histological findings showed a lipoleiomyoma, uterine leiomyoma, and hemangioma in the uterine corpus tissue (Appendix A). There were no findings suggestive of malignancy in the endometrium, cervix, bilateral fallopian tubes, or ovary. Our molecular pathological studies to date have identified five factors (caveolin, cyclin B, cyclin E, LMP2/b1i, and Ki-67) as differentiating markers between uterine leiomyosarcoma and other uterine mesenchymal tumors [8] (Figure 1). Therefore, the expression status of five candidate factors as biomarkers for distinguishing between uterine leiomyosarcoma and other uterine mesenchymal tumors was examined in the tumor tissue from this case and uterine leiomyosarcoma tissue.

The expression of caveolin, a molecular biomarker for uterine mesenchymal tumors, was clearly confirmed in uterine leiomyoma, uterine leiomyosarcoma, and our patient’s uterine tumor, i.e., lipoleiomyoma (Figure 1 and Figure 2). Mild expression of cyclin B, which is considered a malignant mesenchymal tumor biomarker, was confirmed in uterine leiomyosarcoma and our patient’s uterine tumor (i.e., lipoleiomyoma) (Figure 1 and Figure 2). A strong expression of cyclin E and Ki-67, which are candidate biomarkers for malignant uterine mesenchymal tumors, was confirmed in uterine leiomyosarcoma (Figure 1 and Figure 2). Mild expression of cyclin E and Ki-67 was confirmed in the patient’s uterine tumor (i.e., lipoleiomyoma) (Figure 1 and Figure 2). Previous research reports revealed that mice lacking LMP2/β1i, one of the immunoproteasome sub-unit factors, develop spontaneous uterine leiomyosarcoma [9]. Recent reports demonstrated that, in human uterine leiomyosarcoma, expression of LMP2/β1i is markedly reduced [9,10]. A strong expression of LMP2/β1i was found in our patient’s uterine tumor (i.e., lipoleiomyoma) similar to normal myometrium tissue and uterine leiomyoma (Figure 1 and Figure 2). Based on these results, the patient’s lipoleiomyoma was probably a benign tumor; however, the possibility of malignant properties cannot be ruled out.

CD31, CD34, factor VIII, D2-40, etc., are used as vascular tissue biomarkers, including for blood vessels, in the histopathological diagnosis of vascular tumors. Of these biomarkers, CD31 has the highest specificity for vascular tumors because it is specifically expressed on vascular and lymphatic endothelial cells [11,12]. Therefore, on tissues suspected of hemangioma and uterine leiomyosarcoma, immune-histochemical staining was performed using an anti-human CD31 antibody. As a result of the IHC experiment, many CD31-positive cells were observed in the hemangioma-suspected tissue; however, only a few CD31-positive cells were found in the uterine leiomyosarcoma tissue (Figure 3A). CD31-positive cells were found to be statistically higher in suspected hemangioma tissue than in uterine leiomyosarcoma tissue (Figure 3B).

We examined the lipoleiomyoma and hemangioma developing sites in the excised tissue specimens to explore the causal relationship between concurrently occurring lipoleiomyoma and hemangiomas. The findings showed no hemangioma in the lipoleiomyoma tissue, but hemangiomas were observed in the smooth muscle tissue adjacent to the lipoleiomyoma tissue (Figure 4). In the area of hemangioma tissue and lipoleiomyoma tissue in the slides examined for surgical pathology by our medical staff, the content of hemangioma tissue was approximately 2.2–5.3% of the total tumor (Figure 4, Appendix A), while lipoleiomyoma tissue constituted approximately 18.3–36.6%. The remaining tissue consisted of leiomyoma (Figure 4, Appendix A).

## 4. Discussion

Typical leiomyomas, characterized by intersecting short fascicles of bland spindled cells with cigar-shaped nuclei, are most common; however, cellular mitotically active, myxoid, epithelioid, bizarre nuclei, and lipoleiomyoma forms also exist. Lipoleiomyoma has a variable number of mature adipocytes admixed with smooth muscle cells. First described in 1897, uterine hemangioma was an incidental discovery from an autopsy of a young woman who developed anemia and dyspnea and died 24 h after delivering twins [13]. It is a rare benign vascular tumor that can cause bleeding problems in various age groups. In many cases, uterine hemangiomas cause menorrhagia and pregnancy-related complications.

A 54-year-old non-pregnant woman presented to our medical facility with significant vaginal bleeding. An outpatient MRI imaging study revealed a mass measuring 140 × 100 mm, suggestive of a uterine lipoleiomyoma. After surgical treatment, the tumor was submitted for pathological examination, revealing a concurrently occurring uterine lipoleiomyoma and hemangioma. An IHC experiment with appropriate antibodies for CD31 and the five candidate factors as biomarkers demonstrated differential expression of the five candidate factors and CD31, compared with other mesenchymal and leiomyomatoid tumors. Hence, our case was established to be a concurrently occurring uterine lipoleiomyoma and hemangioma. To the best of our knowledge, this is the first case of a concurrent uterine hemangioma and intramural lipoleiomyoma (Table 1, Appendix A). Similar to normal myometrium tissue and a uterine leiomyoma, strong expression of LMP2/β1i was found in the patient’s uterine tumor (i.e., lipoleiomyoma). Through a molecular pathological analysis using the five biomarker candidates currently under investigation, the patient’s tumor was classified as a benign mesenchymal tumor.

Unlike cancer, where malignant tumors originate from epithelial cells, uterine mesenchymal tumor cells have various cell structures, sizes, and nuclear atypia. Therefore, surgical pathological diagnosis of uterine mesenchymal tumors, including benign and/or malignant tumors, is often difficult. As such, in clinical practice, it is necessary to apply the immune-histochemical staining method using a biomarker specific to uterine leiomyosarcoma, a malignant tumor, in order to achieve a more accurate surgical pathological diagnosis. In a joint clinical study (named the PRUM-iBio study) with 28 medical facilities, our medical team examined candidate factors as biomarkers specific to uterine leiomyosarcoma using tissues from a total of 336 cases of uterine mesenchymal tumor, focusing on patients suspected of developing a benign tumor, uterine leiomyoma, patients suspected of developing a malignant tumor, uterine leiomyosarcoma, and patients with other mesenchymal tumors [14].

Lipoleiomyoma also has a variety of cytoskeletal components, making differential diagnosis difficult in some cases. Therefore, our medical staff performed immune-histochemical staining using candidate factors as uterine leiomyosarcoma-specific biomarkers on tissue excised from a patient with lipoleiomyoma. The results showed that, in our case of lipoleiomyoma, the expression pattern of each factor clearly differed from that of uterine leiomyosarcoma and resembled that of uterine leiomyoma. Furthermore, the results of our clinical study demonstrated that MRI imaging tests for uterine mesenchymal tumors have no differential diagnostic efficacy for tumor malignancy. In other words, there are limitations regarding the differentiation between uterine myoma and sarcoma using MRI examination [20]. MRI imaging for uterine mesenchymal tumors is an essential preoperative test to determine the mass size and location. To establish a more accurate histopathological diagnosis for uterine mesenchymal tumors with diverse cellular presentation, the tissue staining results obtained from our lipoleiomyoma cases provide useful molecular pathological information to determine treatment strategies.

It is necessary to consider whether uterine-derived lipoleiomyoma and soft tissue-derived lipoleiomyoma have similar biological characteristics. Some case reports show that uterine-derived lipoleiomyoma tends to increase after menopause, and it is believed that it may have an estrogen-independent proliferative ability [21,22]. Conversely, several women are diagnosed with lipoleiomyoma with irregular menstruation as the chief complaint. Previous studies using isolated human tissues revealed that uterine mesenchymal stem cells are present between the endometrium and the smooth muscle layer junction. It is believed that reactive oxygen generated by repeated ischemia and reperfusion of blood in the uterine smooth muscle layer during the monthly menstrual cycle causes genetic abnormalities in uterine mesenchymal stem cells, producing leiomyoma cells, i.e., uterine mesenchymal stem cells resulting in leiomyoma cells. In this patient, adipocytes are thought to have proliferated in the leiomyoma tissue for some reason, but the cause of uterine lipoleiomyoma was not confirmed. The results obtained from contrast-enhanced MRI provide positive clinical information for differentiation. In contrast, the prevalence of uterine hemangioma is high in patients with liver cirrhosis and those taking oral contraceptives. Furthermore, uterine hemangiomas are thought to develop due to abnormal secretion of estrogen. Hemangioma-like abnormalities that develop during pregnancy or during oral contraceptive use resolve spontaneously 6–9 months after delivery or discontinuation of oral contraceptives. However, a significant number of patients with uterine hemangiomas present with abnormal vaginal bleeding [23], which was demonstrated in our patient. Since vaginal bleeding is difficult to control with medical management, it is important to consider the patient’s intention to conceive and select the appropriate surgical treatment for the uterine hemangioma.

Recent clinical research revealed that Germline *Von Hippel-Lindau* (*VHL*) pathogenic mutations have been implicated in the development of vascular benign tumors, including hemangiomas and hemangioblastomas [24]. However, VHL pathogenic mutations have not been reported to be involved in the development of uterine leiomyoma, lipoleiomyoma, or uterine leiomyosarcoma. In other words, the origin cells that differentiate into hemangiomas are considered to be genetically different from uterine leiomyoma cells, lipoleiomyoma cells, and uterine leiomyosarcoma cells. Therefore, it is natural that there is a significantly higher rate of CD31-positive cells in suspected hemangiomas than in uterine leiomyosarcoma tissues (Figure 3). Furthermore, it is believed that no hemangiomas are found in lipoleiomyoma tissue due to genetic differences between different tumor cells (Figure 4, Appendix A).

In cases where the development of a tumor derived from the uterine smooth muscle layer is suspected based on the imaging test results and if the tumor grows rapidly or abnormal bleeding is observed, periodic follow-up and confirmation of the presence or absence of malignant findings should be repeatedly performed. The limitations of preoperative diagnosis and the economic and psychological burden on patients and their families due to repeated tests should also be considered. In the future, more cases are necessary for data analysis, including cases where verification or follow-up was performed for uterine lipoleiomyomas and uterine hemangiomas at well-experienced medical facilities. The best treatment for hemangioma has not been established despite remarkable advances in medical technology. However, accurate clinical diagnosis is critical to prevent the overtreatment of women of reproductive age. Although the prognosis is excellent after hysterectomy, further basic medical research and clinical research are needed to establish a cure for uterine hemangiomas.

## 5. Conclusions

Hemangiomas, or benign tumors of the blood vessels, rarely occur in the uterus. However, they can cause many clinical complications, including abdominal pain and excessive vaginal bleeding. To the best of our knowledge, our case is the first patient reported to have a concurrent uterine hemangioma and lipoleiomyoma. The possibility of several types of mesenchymal tumors must be considered in the differential diagnosis of patients with abnormal vaginal bleeding. Therefore, pathological examination by endometrial aspiration biopsy and cervical cytology, and close monitoring of the MRI results should be conducted by the medical staff with consideration for the patient’s desire for pregnancy, including surgical treatment options for uterine hemangioma.

## Figures and Tables

**Figure 1 diagnostics-12-02468-f001:**
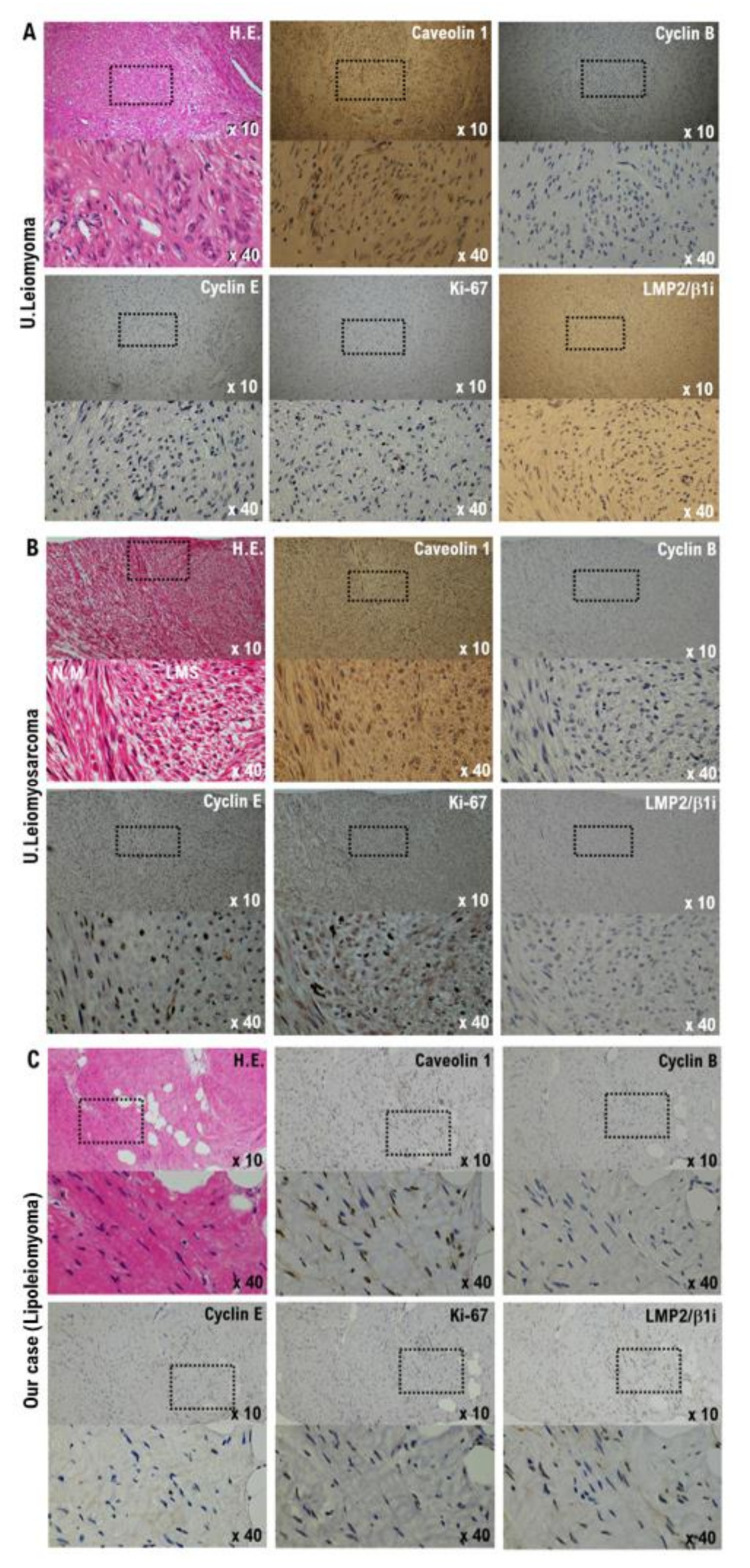
Differential expression of cyclin B, cyclin E, caveolin 1, ki-67, and LMP2/β1i as potential biomarkers in the normal myometrium, uterine leiomyoma, uterine leiomyosarcoma, and uterine tumor in our cases. (**A**) The image shows a spindle cell leiomyoma. The low-power view (10× field) shows a well-circumscribed tumor nodule in the myometrium composed of broad fascicles of spindle cells. The high-power view (40× field) shows a uterine leiomyoma (spindle cell) with bland cytological features, elongated nuclei, and fine nuclear chromatin. Immuno-histochemistry (IHC) of uterine leiomyoma tissue sections was performed using monoclonal antibodies. (**B**) The image shows uterine epithelioid leiomyosarcoma. The low-power view (10× field) shows a uterine mass and an irregular interface with the myometrium composed of round to polygonal cells with granular eosinophilic cytoplasm. Significant nuclear atypia and mitoses are evident. The ligh-power view (40× field) shows tumor cells that are round to ovoid. The tumor cells have eosinophilic granular cytoplasm and irregularly shaped nuclei. IHC of the leiomyosarcoma tissue sections was performed using the appropriate monoclonal antibodies. (**C**) An admixture of round, polygonal, bizarre, or spindle cells, with marked atypia, with or without giant cells, and fat sprouts-like cells was seen in the uterine tumor in our cases. Some tumors invaded the lymphatic vessels. The low-power view (10× field) shows no obvious high-grade nuclear atypia or mitotic cell proliferation, and necrosis is observed. A high-power view (40× field) showing tumor cells with significant pleomorphism, whereas some are multi-nucleated, and lipomyoblastic differentiation is evident. Using the appropriate monoclonal antibodies, IHC of the normal the myometrium, leiomyoma, leiomyosarcoma, and uterine tumor was performed in our cases. Tissues derived from the patient with concurrent leiomyoma and leiomyosarcoma were used for IHC examination. Details are provided in the Appendix A.

**Figure 2 diagnostics-12-02468-f002:**
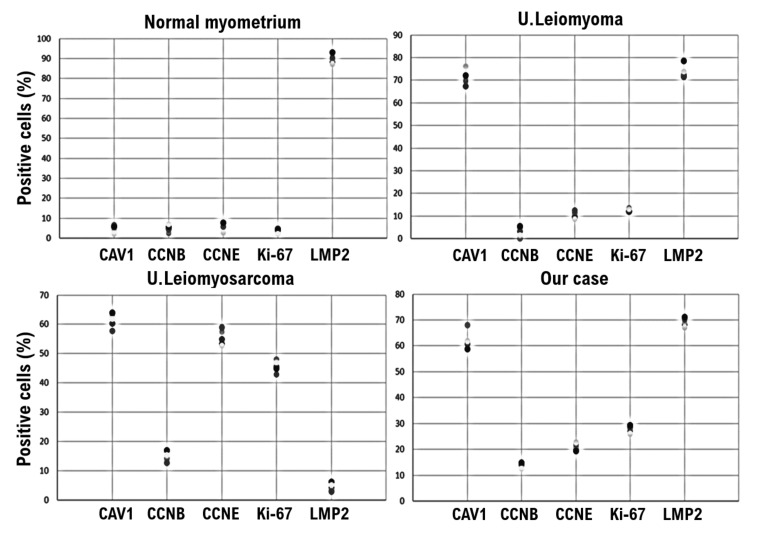
Cyclin E and Ki-67-slightly positive uterine mesenchymal tumor cells in the tumor in our case compared with a normal myometrium and uterine leiomyoma. IHC of the normal myometrium, uterine leiomyoma, uterine leiomyosarcoma, and the uterine tumor tissues in our case (i.e., uterine lipoleiomyoma) was performed using appropriate monoclonal antibodies. The tissues were randomly selected from the normal myometrium, uterine leiomyoma, uterine leiomyosarcoma, and the uterine tumor from our case. Under a 40× field of view, the rates of positivity for the five biomarkers were determined in five tissue sites under a microscope (Panthera Shimadzu Co. Ltd., Kyoto, Japan). The positivity rate of cells at five different locations in the tissue is shown as dots with a five-color gradation from white to black. The positive rates at each site for each tissue are displayed in a scatter plot. CAV1: caveolin 1; CCNB: cyclin B; CCNE: cyclin E; LMP2: LMP2/β1i. U.leiomyoma: uterine leiomyoma; U.leiomyosarcoma: uterine leiomyosarcoma; U.lipoleiomyoma: uterine lipoliomyoma.

**Figure 3 diagnostics-12-02468-f003:**
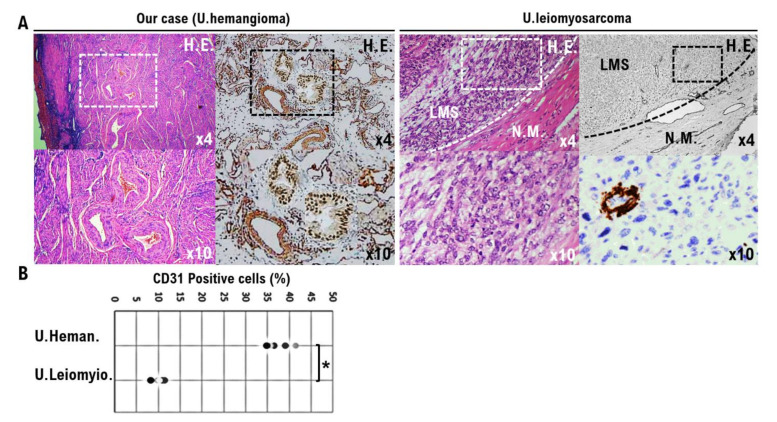
Significance of CD31-positive uterine hemangioma cells in our cases. Differential expression of CD31 as a biomarker for vessel endothelial cells, such as vascular endothelial cells and lymphatic endothelial cells, was observed in the hemangioma from our case. (**A**) The photograph shows the hemangioma in our case, uterine leiomyosarcoma, and normal myometrium. The low-power view (10× field) shows the uterine mass’ irregular interface with normal myometrium, which is composed of round to polygonal cells with granular eosinophilic cytoplasm. Significant nuclear atypia and mitoses are evident. A high-power view (10× field) shows round to ovoid tumor cells. The tumor cells have an eosinophilic granular cytoplasm and irregularly shaped nuclei. IHC was performed with the leiomyosarcoma tissue sections using the appropriate monoclonal antibodies (left upper panel). The uterine tumor in our case exhibits capillary-sized vessels that are lined by plump endothelial cells. The low-power view (4× field) shows no obvious high-grade nuclear atypia or mitotic cell proliferation. The high-power view (10× field) shows that vessel endothelial cells exhibit significant pleomorphism, and some show multi-nucleated, vessel endothelial differentiation. Using the appropriate monoclonal antibodies, IHC was performed with the hemangioma tissue from our case, leiomyosarcoma tissue, and normal myometrium (upper right panel). (**B**) The five tissue sites were randomly selected from normal myometrium, uterine leiomyosarcoma, and the hemangioma tissue from our case. In the 10× field of view, the positive rates for the biomarker were determined in three tissue sites under a microscope (Panthera Shimadzu Co. Ltd., Kyoto, Japan) (lower panel). The positive rates at the sites for each tissue are shown in a scatter plot. Statistical significance; * *p* < 0.001.

**Figure 4 diagnostics-12-02468-f004:**
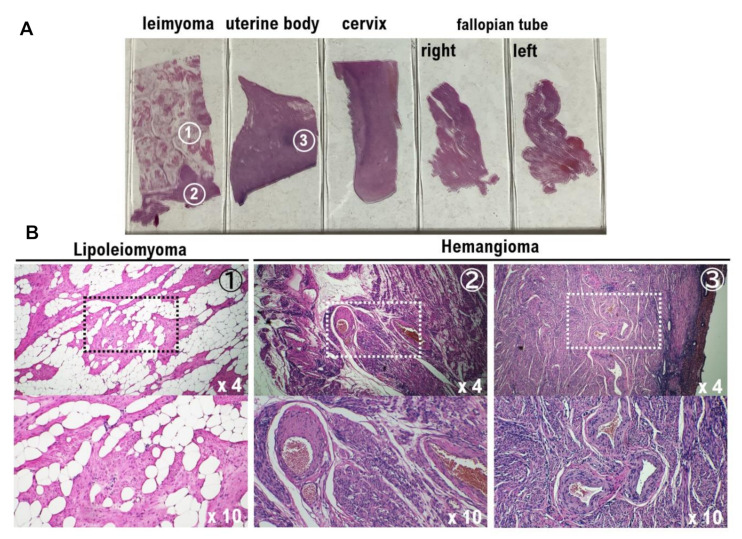
Identification of lipoleiomyoma and hemangioma locations by histological and pathological examination. (**A**) Upper panel: block of excised tissue fixed by formalin; bottom panel: microscopic histology of each excised tissue block. (**B**) Uterine lipoleiomyoma: a mixture of mature adipocytes was observed within the uterine leiomyoma. Uterine hemangioma: uterine hemangiomas consisting of irregularly shaped cavernous spaces infiltrating between the myometrial fascicles. The vascular spaces are lined by flattened endothelium (positive for CD31) and distended with blood. No hemangioma was observed in the uterine lipoleiomyoma tissue; however, hemangiomas were observed in the smooth muscle tissue adjacent to the uterine lipoleiomyoma tissue.

**Table 1 diagnostics-12-02468-t001:** Differential expressions of SMA, caveolin1, cyclin B, cyclin E, LMP2, NT5DC2, CD133, and Ki-67 in human uterine mesenchymal tumors and uterine LANT-like tumor.

Mesenchymal Tumor Types	Age Years	n	Protein Expression *							
			SMA	CAV1	CCNB	CCNE	LMP2	NT5DC2	CD133	Ki-67
**Normal**	30–80s	74	+++	−	−	−	+++	−	−	−
**Leiomyoma (LMA)** (Ordinally leiomyoma) (Cellular leiomyoma)	30–80s	**40** (30)(10)	+++	++	−/+	−/(+)	+++	−/+	−	+/−
			+++	++	−/+	−	+++	−/+		+/−
			++	++	−/+	−/(+)	++	−/+		+/−
**STUMP**	40–60s	**12**	++	++	+	−/+	−/+	−/+	NA	+/+++
**Lipoleiomyoma**	40–50s	**2**	NA	++	−/+	+	+++	NA	NA	++
**Bizarre Leiomyoma**	40–50s	**4**	++	++	−/+	+	Focal+	+	NA	+
**Intravenous LMA**	50s	**3**	++	++	+	+	−	NA	++	+
**Benign metastasizing**	50s	**1**	++	++	+	++	−	NA	NA	++
**Leiomyosarcoma**	30–80s	**54**	−/+	+	++	+++	−/+	++	++	++/+++
**Rhabdomyosarcoma**	10s, 50s	**2**	NA	++	−/+	+++	+++	NA	NA	NA
**U.LANT** ** ^#^ ** **-like tumor**	40s	**1**	++	+	NA	++	−	NA	NA	−

* Staining score of expression of SMA, CAV1 (caveorin 1), CCNB (cyclin B), CCNE (cyclin E), LMP2 (low molecular protein 2), NT5DC2 (5’-nucleotidase domain-containing 2), and Ki-67 from results of IHC experiments. Protein expression *: estimated protein expressions by immunoblot analysis, immune-histochemistry (IHC) and/or RT-PCR (quantitative-PCR); −/+: partially positive (5% to 10% of cells stained); focal+: focal-positive (focal or sporadic staining with less than 5% of cells stained); ++: staining with 5% or more, less than 90% of cells stained; +++: diffuse-positive (homogenecus distribution with more than 90% of cells stained); −: negative (no stained cells). U.LANT-like tumor: uterine leiomyomatoid angiomatous neuroendocrine tumor-like tumor, LMP2 [14,15], cyclin E [14,15], caveolin1 [15] NT5DC2 [16,17], CD133 [16] Ki-67 [14,15]. STUMP (smooth muscle tumor of uncertain malignant potential) [16,18]. Cyclin E, LMP2, caveolin1 are potential biomarker for human uterine mesenchymal tumors. LANT^#^, leiomyomatoid angiomatous neuroendocrin tumor (LANT) is described as a dimorphic neurosecretory tumor with a leiomyomatous vascular component [19]. NA: no answer.

## Data Availability

The study did not report any data.

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
