# Peer review of "Characteristic of Concurrent Uterine Lipoleiomyoma and Hemangioma by Algorithm of Candidate Biomarkers for Uterine Mesenchymal Tumor"

_diagnostics, 2022, doi:10.3390/diagnostics12102468_

Round 1

Reviewer 1 Report

(1)   Figure 3 and 4 were put in wrong order. The Figure 3 was actually Figure 4, and the Figure 4 was actually Figure 3.

(2)   Were there gross picture and MRI image of the uterine tumor?

(3)   What is the percentage of hemangioma versus lipoleiomyoma in volume in this case?

(4)   In page 10, the “Conclusion” did not correlate with the main objective of the manuscript. As mentioned in the abstract, the abstract had better be similar to that in the abstract “As far as we know, this case is the first patient of concurrent uterine hemangioma and lipoleiomyoma. Hence, such occurrence of several types of mesenchymal tumors”.

(5)   In the abstract, “on tissue excised from patients with uterine leiomyoma.” Had better be changed to “on tissue excised from a patient with uterine tumor”.

(6)   In the "Introduction" section, please describe if uterine hemangioma belong to vascular malformation or not.

(7)   In the “Results” section, the title of “case 1” is inappropriate because there was only one case in the manuscript.

(8)   The term “case 1” in the manuscript might be changed to “the index case” or “our case”, etc.

(9)   In Figure 2, the “U. Lipoleiomyoma” should be marked with “(our case)” or else.

(10)   In the first paragraph of the “Results” section, was “uterine cytology” endometrial biopsy or cervical cytology?

(11)   In the real Figure 4 (i.e., the current Figure 3), why was the CD31 stain so faint?

(12)   Page 2, line 31: “intramural leiomyoma” had better be “intramural lipoleiomyoma”.

(13)   Page 2, lines 43-44:  “Our clinical research group purchased monoclonal anti-bodies against CD31 (GTX130274) was purchased from GeneTex, Inc. (Irvine, CA, USA).” should be re-written.

(14)   Page 6, line 8, 11: “and patient’s” had better be “and our patient’s”.

Author Response

Manuscript ID: diagnostics-1898281

Reviewer 1

Comments and Suggestions for Authors

Comment 1 Figure 3 and 4 were put in wrong order. The Figure 3 was actually Figure 4, and the Figure 4 was actually Figure 3.

Answer 1 We appreciate and agree with your comment. This mistake in the figure order was caused by the editorial department. Accordingly, we changed the order of these figures in the revised manuscript.

Comment 2 Were there gross picture and MRI image of the uterine tumor?

Answer 2 We appreciate your comment. The gross picture and MRI image of the uterine tumor are indicated in the supplementary information as follows.

Supplementary Figure 1. Contrast images of magnetic resonance imaging (MRI)

Contrast MRI images, T2 BLADE Tra and T2 TES Sag, clearly show the mass in our case (patient with uterine tumor). The white dotted circle indicates the uterine tumor.

Supplementary Figure 2. Gross and histopathological morphology of the uterine mesenchymal and vascular tumors. A. Gross findings of formalin-fixed excised tissue. The black discolored tissues contain many capillaries. Hemangioma develops and proliferates in these tissues featuring substantial blood flow. B. At the uterine body, gray-white masses are observed (upper masses), including the leiomyoma tissue. No evidence of benign or malignant tumors is expected in the cervical tissue block. In bilateral fallopian tubes and ovaries, dark discolored tissue is observed. Normal fallopian tubes and ovarian tissue have many capillaries, explaining the prominent blood flow. Therefore, black discoloration is seen in the fallopian tubes and ovaries. C. The cell tissue findings were examined in each block where the excised tissue was cut into 11 sections. The results show lipoleiomyoma and hemangioma in the tissue contained in the fourth block from the top. D. From the histopathological findings of the 11 sections of excised tissue, it is believed that the hemangioma did not grow inside the lipoleiomyoma. Rather, the hemangioma grew in the vicinity of the lipoleiomyoma-affected tissue.

Comment 3 What is the percentage of hemangioma versus lipoleiomyoma in volume in this case?

Answer 3 We appreciate your comment. This issue is important for investigating the mechanism by which hemangioma and lipoleiomyoma co-occur. However, it is not easy to give an accurate reply to this question. We make the following comment in response to this comment.

In the area of hemangioma tissue and lipoleiomyoma tissue observed in the tissue sections of the slides examined for surgical pathology by our medical staff, the content of hemangioma tissue was approximately 2.2%–5.3% of the total tumor (Figure 4, Supplementary Figure 2), while lipoleiomyoma tissue constituted approximately 18.3%–36.6%. The remaining tissue consisted of leiomyoma (Figure 4, Supplementary Figure 2).

Comment 4 In page 10, the “Conclusion” did not correlate with the main objective of the manuscript. As mentioned in the abstract, the abstract had better be similar to that in the abstract “As far as we know, this case is the first patient of concurrent uterine hemangioma and lipoleiomyoma. Hence, such occurrence of several types of mesenchymal tumors”.

Answer 4 We appreciate and agree with this comment. Accordingly, we rewrote the Conclusion in the revised manuscript as follows.

Conclusion in Abstract

To the best of our knowledge, our case is the first reported case of concurrent uterine hemangioma and lipoleiomyoma. Hence, such occurrence of several types of mesenchymal tumors must be considered in the differential diagnosis of patients with abnormal vaginal bleeding. As such, molecular pathological examination and close monitoring of the MRI results should be conducted by medical staff while considering the patient’s desire for pregnancy, including surgical treatment options for uterine hemangioma.

Conclusion

Hemangiomas, or benign tumors of the blood vessels, rarely occur in the uterus. However, they can cause many clinical complications, including abdominal pain and excessive vaginal bleeding. To the best of our knowledge, our case is the first patient reported to have concurrent uterine hemangioma and lipoleiomyoma. Such occurrence of several types of mesenchymal tumors must be considered in the differential diagnosis of patients with abnormal vaginal bleeding. Therefore, pathological examination by endometrial aspiration biopsy and cervical cytology, and close monitoring of the MRI results should be conducted by medical staff while considering the patient’s desire for pregnancy, including surgical treatment options for uterine hemangioma.

Comment 5 In the abstract, “on tissue excised from patients with uterine leiomyoma.” Had better

be changed to “on tissue excised from a patient with uterine tumor”.

Answer 5 We appreciate and agree with this comment. We changed the text accordingly.

Comment 6 In the "Introduction" section, please describe if uterine hemangioma belong to vascular malformation or not.

Answer 6 We appreciate this comment. Accordingly, we added a comment regarding the histopathological differences between uterine hemangioma and vascular malformation as follows.

Hemangiomas and vascular malformations on the body surface and soft tissue are often commonly referred to as “hemangioma.” However, hemangioma and vascular malformation are different diseases according to the International Society for the Study of Vascular Anomalies (ISSVA) classification, which is being internationally standardized and advocated by the International Society for the Treatment of Hemangiomas and Vascular Malformations (4).

  1. Wassef M, Blei F, Adams D, et al. Vascular anomalies classification: recommendations from the International Society for the Study of Vascular Anomalies. Pediatrics. 2015; 136:e203–214.

Comment 7 In the “Results” section, the title of “case 1” is inappropriate because there was only one case in the manuscript.

Answer 7 We appreciate and agree with this comment. Accordingly, we changed the text from “Case 1” to “our case.”

Comment 8 The term “case 1” in the manuscript might be changed to “the index case” or “our case”, etc.

Answer 8 We appreciate and agree with this comment. Accordingly, we changed the text from “Case 1” to “our case.”

Comment 9 In Figure 2, the “U. Lipoleiomyoma” should be marked with “(our case)” or else.

Answer 9 We appreciate and agree with this comment. Accordingly, we changed “U. Lipoleiomyoma” to “(Our case)” in Figures 1C, 2, and 3A.

Comment 10 In the first paragraph of the “Results” section, was “uterine cytology” endometrial biopsy or cervical cytology?

Answer 10 We appreciate this comment. “Uterine cytology” involves pathological examination by endometrial aspiration biopsy and pathological diagnosis by cervical cytology. Thank you for pointing this issue out. We have amended the text about the tests that were performed.

Comment 11 In the real Figure 4 (i.e., the current Figure 3), why was the CD31 stain so faint?

Answer 11 We appreciate your comment.

When we performed microscopic examination of slides subjected to immunohistochemical staining, CD31 staining was intense. However, as noted by the reviewer, the CD31 staining is rather faint. We photographed the tissue in the same area. We have replaced the photo with a new one for clarity.

Comment 12 Page 2, line 31: “intramural leiomyoma” had better be “intramural lipoleiomyoma”.

Answer 12 We appreciate and agree with this comment. We changed the text accordingly.

Comment 13 Page 2, lines 43-44: “Our clinical research group purchased monoclonal anti-bodies against CD31 (GTX130274) was purchased from GeneTex, Inc. (Irvine, CA, USA).” should be re-written.

Answer 13 We appreciate and agree with this comment. We rewrote the text as follows.

Our clinical research group purchased monoclonal antibodies against CD31 (GTX130274) was purchased from GeneTex, Inc. (Irvine, CA, USA).

Comment 14 Page 6, line 8, 11: “and patient’s” had better be “and our patient’s”.

Answer 14 We appreciate and agree with this comment. We rewrote the text accordingly.

Reviewer 2 Report

This is the first case report of a coexisting uterine lipoma and uterine hemangioma.

In this case, the patient underwent a total hysterectomy, and the subsequent pathological examination revealed various findings.

The immunostaining itself may be useful in confirming the diagnosis.

We feel obligated to report this unusual case for the benefit of the next person who experiences it.

However, the paper has the shortcomings listed below. Areas for improvement and areas of difficulty in understanding are described below.

Although a different case from the present case is shown as an example in Figures, this is not properly indicated in the text or in the figure legend, which makes it difficult for the reader to understand.

The discussion of immunostaining is not sufficient to describe the significance and usefulness of immunostaining.

In Figure 1A and B, if leiomyoma or sarcoma is included as a contrast to case 1, the source of the case should be clearly indicated.

In Figure 2, it is not clear what the black and white dots are.

Figure 3 and Figure 4 should be summarized and concise.

Reports of uterine lipomas and uterine hemangiomas themselves are commonly encountered in daily practice and are not particularly uncommon.

This immunostaining is possible only as a result of a total hysterectomy as treatment, and preoperative evaluation, such as MRI, is considered essential for differentiation.

The results of this immunostaining are not useful in deciding the treatment strategy, and the diagnostic significance of the immunostaining is the main focus of the paper, but the significance of the immunostaining is not well described in the text.

Therefore, the paper does not provide much useful information.

Author Response

Manuscript ID: diagnostics-1898281

Reviewer 2

Comments and Suggestions for Authors

This is the first case report of a coexisting uterine lipoma and uterine hemangioma.

In this case, the patient underwent a total hysterectomy, and the subsequent pathological examination revealed various findings.

The immunostaining itself may be useful in confirming the diagnosis.

We feel obligated to report this unusual case for the benefit of the next person who experiences it.

However, the paper has the shortcomings listed below. Areas for improvement and areas of difficulty in understanding are described below.

Comment 1 Although a different case from the present case is shown as an example in Figures, this is not properly indicated in the text or in the figure legend, which makes it difficult for the reader to understand.

Answer 1 We appreciate and agree with this comment. Accordingly, we added the details of our patient with co-occurring uterine leiomyoma and leiomyosarcoma in the revised manuscript and Supplementary information as follows.

Regarding the photographs of tissues subjected to immunohistochemical staining using uterine leiomyoma and uterine leiomyosarcoma tumor tissues as shown in Figure 1A and 1B, this uterine tumor was excised from a patient suspected of developing uterine leiomyosarcoma by our medical team. In many patients, uterine leiomyosarcoma co-occurs with uterine leiomyoma. Therefore, uterine leiomyoma and uterine leiomyosarcoma tumor tissues were surgically excised from the same patient.

Age: 70s, Sex: Female

In April 2019: We performed surgical treatment for patients suspected of developing uterine leiomyosarcoma: abdominal simple total hysterectomy, bilateral appendectomy, reticulometry of the reticulum, and mesentero-disseminated lesion resection.

May–October 2019: Combination therapy with docetaxel (DTX) and gemcitabine (GEM) was initiated (six cycles in total)

October 2021: PET-CT scan examination revealed recurrence of the tumor in the pelvis.

Treatment with pazopanib was initiated.

January 2022: CT scan to examine the therapeutic efficacy of pazopanib revealed PD.

February 2022: Treatment with doxorubicin (DXR) was initiated.

July 2022: At the time of writing, the eighth cycle has ended.

By CT, the therapeutic effect of DXR was evaluated as SD.

Histopathological diagnosis: Concurrence of uterine leiomyoma and uterine leiomyosarcoma is recognized.

Comment 2 The discussion of immunostaining is not sufficient to describe the significance and usefulness of immunostaining.

Answer 2 We appreciate and agree with this comment. Accordingly, we added the importance and efficacy of immunostaining as supplementary medical evidence for a surgical pathological diagnosis as follows.

Unlike cancer, where malignant tumors originate from epithelial cells, uterine mesenchymal tumor cells have various cell structures, sizes, and nuclear atypia. Therefore, surgical pathological diagnosis of uterine mesenchymal tumors, including benign and/or malignant tumors, is often difficult. As such, in clinical practice, it is desired to establish an immunohistochemical staining method using a biomarker specific to uterine leiomyosarcoma, a malignant tumor, in order to achieve more accurate surgical pathological diagnosis. In a joint clinical study (named the PRUM-iBio study) with 28 medical facilities, our medical team is examining candidate factors as biomarkers specific to uterine leiomyosarcoma using tissues from a total of 336 cases of uterine mesenchymal tumor, focusing on patients suspected of developing benign tumor, uterine leiomyoma, patients suspected of developing malignant tumor, uterine leiomyosarcoma, and patients with other mesenchymal tumors (20).

Lipoleiomyoma also has a variety of cytoskeletal components, making differential diagnosis difficult in some cases. Therefore, our medical staff performed immunohistochemical staining using candidate factors as uterine leiomyosarcoma-specific biomarkers on tissue excised from our patient with lipoleiomyoma.

The results showed that, in our case of lipoleiomyoma, the expression pattern of each factor clearly differed from that of uterine leiomyosarcoma and resembled that of uterine leiomyoma. Furthermore, the results of our clinical study demonstrated that MRI imaging tests for uterine mesenchymal tumors have no differential diagnostic efficacy for tumor malignancy. In other words, there are limitations regarding the differentiation between uterine myoma and sarcoma using MRI examination (21). MRI imaging for uterine mesenchymal tumors is an essential preoperative test to determine mass size and location. To establish a more accurate histopathological diagnosis for uterine mesenchymal tumors with diverse cellular presentation, results of staining of tissues obtained from our cases with lipoleiomyoma provide useful molecular pathological information to determine treatment strategies.

  1. https://upload.umin.ac.jp/cgi-open-bin/ctr/ctr_view.cgi?recptno=R000044182
  2. Suzuki A, Aoki M, Miyagawa C, et al. Differential Diagnosis of Uterine Leiomyoma and Uterine Sarcoma Using Magnetic Resonance Images: A Literature Review. Healthcare 2019, 7(4), 158; https://doi.org/10.3390/healthcare7040158

Comment 3 In Figure 1A and B, if leiomyoma or sarcoma is included as a contrast to case 1, the source of the case should be clearly indicated.

Answer 3 We appreciate and agree with this comment. Accordingly, we added the details of our patient with co-occurring uterine leiomyoma and leiomyosarcoma in the revised manuscript and Supplementary information as follows.

Regarding the photographs of tissues subjected to immunohistochemical staining using uterine leiomyoma and uterine leiomyosarcoma tumor tissues in Figure 1A and 1B, this uterine tumor was excised from a patient suspected of developing uterine leiomyosarcoma by our medical team. In many patients, uterine leiomyosarcoma co-occurs with uterine leiomyoma. Therefore, uterine leiomyoma and uterine leiomyosarcoma tumor tissues were surgically excised from the same patient.

Age: 70s, Sex: Female

In April 2019: We performed surgical treatment for patients suspected of developing uterine leiomyosarcoma: abdominal simple total hysterectomy, bilateral appendectomy, reticulometry of the reticulum, and mesentero-disseminated lesion resection.

May–October 2019: Combination therapy with DTX and GEM was initiated (six cycles in total).

October 2021: PET-CT scan examination revealed recurrence of the tumor in the pelvis.

Treatment with pazopanib was initiated.

January 2022: CT scan to examine the therapeutic efficacy of pazopanib revealed PD.

February 2022: Treatment with DXR was initiated.

July 2022: At the time of writing, the eighth cycle has ended.

By CT, the therapeutic effect of DXR was evaluated as SD.

Histopathological diagnosis: Concurrence of uterine leiomyoma and uterine leiomyosarcoma is recognized.

Comment 4 In Figure 2, it is not clear what the black and white dots are.

Answer 4 We appreciate and agree with this comment. Accordingly, we added the details of the black and white dots in the legend of Figure 2 as follows.

Under a field of view with 40× magnification, the rates of positivity for the five biomarkers were determined in five tissue sites under a microscope (Panthera Shimadzu Co. Ltd., Kyoto, Japan). The positivity rate of cells at five different locations in the tissue is shown as dots with a five-color gradation from white to black.

Comment 5 Figure 3 and Figure 4 should be summarized and concise.

Answer 5 We appreciate your comment. Figure 3 and 4 were put in wrong order. The Figure 3 was actually Figure 4, and the Figure 4 was actually Figure 3. This mistake in the figure order was caused by the editorial department. Accordingly, we changed the order of these figures in the revised manuscript.

Recent clinical research revealed that Germline Von Hippel-Lindau (VHL) pathogenic mutations have been implicated in the development of vascular benign tumors, including hemangiomas and hemangioblastomas (25). However, VHL pathogenic mutations have not been reported to be involved in the development of uterine leiomyoma, lipoleiomyoma, or uterine leiomyosarcoma. In other words, the origin cells that differentiate into hemangiomas are considered to be genetically different from uterine leiomyoma cells, lipoleiomyoma cells, and uterine leiomyosarcoma cells. Therefore, it is natural that there is a significantly higher rate of CD31-positive cells in suspected hemangiomas than in uterine leiomyosarcoma tissues (Figure 3). Furthermore, it is believed that no hemangiomas are found in lipoleiomyoma tissue due to genetic differences between different tumor cells (Figure 4, Supplementary Figure 2).

  1. Von Hippel-Lindau syndrome. Female Genital Tumours, WHO Classification of Tumours, 5th ed., Vol.4. WHO Classification of Tumours Editorial Board. WORLD HEALTH ORGANIZATION. 2020; pp560.

Comment 6 Reports of uterine lipomas and uterine hemangiomas themselves are commonly encountered in daily practice and are not particularly uncommon.

Answer 6 We appreciate and agree with this comment. We also think that uterine lipomas and uterine hemangiomas themselves are commonly encountered in daily practice and are not particularly uncommon. As the reviewer also states, our patient is the first case report of a coexisting uterine lipoma and uterine hemangioma. In our case, the patient underwent a total hysterectomy, and the subsequent pathological examination revealed various findings.

Comment 7 This immunostaining is possible only as a result of a total hysterectomy as treatment, and preoperative evaluation, such as MRI, is considered essential for differentiation.

Answer 7 We appreciate and agree with this comment.

So far, in a joint clinical study with 28 medical facilities (named PRUM-iBio study), our medical team has performed preoperative evaluation such as MRI imaging for a total of 336 cases of uterine mesenchymal tumors, focusing on patients suspected of developing benign tumor (uterine leiomyoma) and patients suspected of developing malignant tumor (uterine leiomyoma). In cases where MRI imaging tests suggest the development of uterine leiomyoma, the development of uterine leiomyosarcoma was confirmed by postoperative surgical pathological examination. In patients with uterine leiomyosarcoma suspected by MRI imaging tests, pathological malignant findings may not be observed by postoperative surgical pathology. The results of our clinical study demonstrated that MRI imaging tests for uterine mesenchymal tumors have no differential diagnostic efficacy for tumor malignancy. In other words, there are limitations regarding the differentiation between uterine myoma and sarcoma using MRI (21). MRI imaging for uterine mesenchymal tumors is an essential preoperative test to determine mass size and location.

  1. Suzuki A, Aoki M, Miyagawa C, et al. Differential Diagnosis of Uterine Leiomyoma and Uterine Sarcoma Using Magnetic Resonance Images: A Literature Review. Healthcare 2019, 7(4), 158; https://doi.org/10.3390/healthcare7040158

Comment 8 The results of this immunostaining are not useful in deciding the treatment strategy, and the diagnostic significance of the immunostaining is the main focus of the paper, but the significance of the immunostaining is not well described in the text. Therefore, the paper does not provide much useful information.

Answer 8 We appreciate and agree with this comment. Accordingly, we added the importance and efficacy of immunostaining as supplementary medical evidence for surgical pathological diagnosis as follows.

Unlike malignant tumors originating from epithelial cells, that is, cancer, uterine mesenchymal tumor cells have various cell structures, sizes, and nuclear atypia. Therefore, surgical pathological diagnosis of uterine mesenchymal tumors, including benign and/or malignant tumors, is often difficult. As such, in clinical practice, it is desired to establish an immunohistochemical staining method using a biomarker specific to uterine leiomyosarcoma, a malignant tumor, in order to achieve more accurate surgical pathological diagnosis. In a joint clinical study (named the PRUM-iBio study) with 28 medical facilities, our medical team is examining candidate factors as biomarkers specific to uterine leiomyosarcoma using tissues from a total of 336 cases of uterine mesenchymal tumor, focusing on patients suspected of developing benign tumor, uterine leiomyoma, patients suspected of developing malignant tumor, uterine leiomyosarcoma, and patients with other mesenchymal tumors (20).

Lipoleiomyoma also has a variety of cytoskeletal components, making differential diagnosis difficult in some cases. Therefore, our medical staff performed immunohistochemical staining using candidate factors as uterine leiomyosarcoma-specific biomarkers on tissue excised from our patient with lipoleiomyoma.

The results showed that, in our case of lipoleiomyoma, the expression pattern of each factor clearly differed from that of uterine leiomyosarcoma and resembled that of uterine leiomyoma. Furthermore, the results of our clinical study demonstrated that MRI imaging tests for uterine mesenchymal tumors have no differential diagnostic efficacy for tumor malignancy. In other words, there are limitations regarding the differentiation between uterine myoma and sarcoma using MRI examination (21). MRI imaging for uterine mesenchymal tumors is an essential preoperative test to determine mass, size, and location. To establish a more accurate histopathological diagnosis for uterine mesenchymal tumors with diverse cellular presentation, results of staining of tissues obtained from our cases with lipoleiomyoma provide useful molecular pathological information to determine treatment strategies.

  1. https://upload.umin.ac.jp/cgi-open-bin/ctr/ctr_view.cgi?recptno=R000044182
  2. Suzuki A, Aoki M, Miyagawa C, et al. Differential Diagnosis of Uterine Leiomyoma and Uterine Sarcoma Using Magnetic Resonance Images: A Literature Review. Healthcare 2019, 7(4), 158; https://doi.org/10.3390/healthcare7040158

Round 2

Reviewer 2 Report

I myself did the peer review and initially had many unfamiliar points and questions.

The revised version is well corrected in that regard.